# NOS2 and COX-2 Co-Expression Promotes Cancer Progression: A Potential Target for Developing Agents to Prevent or Treat Highly Aggressive Breast Cancer

**DOI:** 10.3390/ijms25116103

**Published:** 2024-06-01

**Authors:** Leandro L. Coutinho, Elise L. Femino, Ana L. Gonzalez, Rebecca L. Moffat, William F. Heinz, Robert Y. S. Cheng, Stephen J. Lockett, M. Cristina Rangel, Lisa A. Ridnour, David A. Wink

**Affiliations:** 1Cancer Innovation Laboratory, Center for Cancer Research, National Cancer Institute, National Institutes of Health, Frederick, MD 21702, USA; coutinholl@nih.gov (L.L.C.); elise.femino@nih.gov (E.L.F.); ana.gonzalez@nih.gov (A.L.G.); robert.cheng2@nih.gov (R.Y.S.C.); 2Center for Translational Research in Oncology, ICESP/HC, Faculdade de Medicina da Universidade de São Paulo and Comprehensive Center for Precision Oncology, Universidade de São Paulo, São Paulo 01246-000, SP, Brazil; mcrrangel@gmail.com; 3Optical Microscopy and Analysis Laboratory, Office of Science and Technology Resources, Center for Cancer Research, National Cancer Institute, National Institutes of Health, Frederick, MD 21702, USA; rebecca.moffat@nih.gov; 4Optical Microscopy and Analysis Laboratory, Cancer Research Technology Program, Frederick National Laboratory for Cancer Research, Frederick, MD 21702, USA; will.heinz@nih.gov (W.F.H.); locketts@mail.nih.gov (S.J.L.)

**Keywords:** nitric oxide, biochemistry, breast cancer, therapy

## Abstract

Nitric oxide (NO) and reactive nitrogen species (RNS) exert profound biological impacts dictated by their chemistry. Understanding their spatial distribution is essential for deciphering their roles in diverse biological processes. This review establishes a framework for the chemical biology of NO and RNS, exploring their dynamic reactions within the context of cancer. Concentration-dependent signaling reveals distinctive processes in cancer, with three levels of NO influencing oncogenic properties. In this context, NO plays a crucial role in cancer cell proliferation, metastasis, chemotherapy resistance, and immune suppression. Increased NOS2 expression correlates with poor survival across different tumors, including breast cancer. Additionally, NOS2 can crosstalk with the proinflammatory enzyme cyclooxygenase-2 (COX-2) to promote cancer progression. NOS2 and COX-2 co-expression establishes a positive feed-forward loop, driving immunosuppression and metastasis in estrogen receptor-negative (ER^-^) breast cancer. Spatial evaluation of NOS2 and COX-2 reveals orthogonal expression, suggesting the unique roles of these niches in the tumor microenvironment (TME). NOS2 and COX2 niche formation requires IFN-γ and cytokine-releasing cells. These niches contribute to poor clinical outcomes, emphasizing their role in cancer progression. Strategies to target these markers include direct inhibition, involving pan-inhibitors and selective inhibitors, as well as indirect approaches targeting their induction or downstream effectors. Compounds from cruciferous vegetables are potential candidates for NOS2 and COX-2 inhibition offering therapeutic applications. Thus, understanding the chemical biology of NO and RNS, their spatial distribution, and their implications in cancer progression provides valuable insights for developing targeted therapies and preventive strategies.

## 1. Introduction

The study of the extensive physiologic effects of nitric oxide (NO) has given rise to a substantial biomedical field over the last decades. Its influence extends across numerous physiological and pathological processes: including chronic inflammatory diseases, cancer, cardiovascular diseases, and neurodegeneration. This diatomic radical has been identified as a mediator of both positive and negative effects [1] and its dualistic behavior, often referred to as the “Janus face of NO”, is best comprehended within a chemical context. NO and its downstream reactive nitrogen species (RNS) constitute diffusible gaseous molecules that exert their effects in time-, spatial-, and concentration-dependent manners [2]. Although many recognized pathways involve the interaction with biological macromolecules, the fundamental chemistry of NO/RNS and the modification of atomic targets emerge as the primary influencers of NO-mediated signaling pathways [3]. To provide a comprehensive framework for understanding the target interactions and biological responses associated with these reactive species in various diseases [4,5] we introduce the concept of ‘The Chemical Biology of NO and RNS’. This chemical roadmap offers insights into the concentration-, temporal-, and spatial-dependent mechanisms that impact diverse physiological and pathophysiological states.

The intriguing aspect of NO biology lies in its precise compartmentalization. A notable example is the controlled diffusion of NO from the endothelial layer to the vascular smooth muscle, leading to tissue bed relaxation. This containment is facilitated by the rapid reactivity of NO with erythrocytes and oxyhemoglobin (HbO_2_) [6]. While discussions on the concentration and temporal aspects of NO’s chemical biology have been prevalent, recent attention has turned towards appreciating the subtleties of its spatial considerations [7,8,9,10,11]. While NO donors may generate a uniform NO flux for study in a petri dish, NOS2 expression acts as a localized point source in situ. Consequently, an accurate mechanistic understanding of NO biology necessitates a thorough exploration of its spatial component.

In this review, we examine the principles of chemical biology to elucidate the spatial effects observed in cancer and the tumor microenvironment (TME). We delve into how the reactions of NO and tumor NOS2 and COX-2 localization are intricately intertwined, influencing the progression of numerous cancers toward unfavorable outcomes.

## 2. Principles of NO and RNS Chemical Biology

It is crucial to understand the fundamental chemistry of NO and RNS to elucidate their physical properties and spatial distribution [12]. NO, a diatomic radical molecule, engages in a wide spectrum of chemical reactions that vary based on concentration and location [1,11]. Two primary types of reactions are observed: direct effects, wherein NO binds directly to targets such as hemes in soluble guanylate cyclase (sGC) or the cytochrome P450 (CYP450) family, and other instances involve reactions with HbO_2_ to form nitrate—an essential scavenging mechanism for compartmentalizing NO in specific tissue beds [13,14]. Direct reactions also occur with oxy radicals and Fenton/peroxidase intermediates. Indirect effects encompass those resulting from RNS formed through various NO and nitrite reactions, prominently including S-nitrosation and nitration reactions. Direct mechanisms are associated with low NO fluxes (<100 nM), while indirect mechanisms involve higher NO levels or acidic nitrite-rich areas, such as in phagosomes or lysosomes [11,15] (Figure 1; Figure 2A).

There are additional direct reactions worth considering, such as oxidative stress mechanisms through reactive oxygen species (ROS) [2]. Metal oxo complexes derived from heme proteins, such as peroxidases, or non-heme iron complexes (as seen in the Fenton reaction), can oxidize nitrite to nitrogen dioxide (NO_2_). While hydrogen peroxide (H_2_O_2_) does not directly react with NO, the chemistry associated with Fenton and Haber-Weiss reactions exhibits a robust interaction with this oxygen radical. H_2_O_2_ reacts notably with many heme proteins, especially peroxidases, forming high-valent iron species that subsequently react with NO to produce nitrite (distinct from nitrate, as observed with HbO_2_/Myoglobin). The Fenton/Haber-Weiss cycle is central to the ROS cycle, where peroxide generates high-valent iron oxo species, such as ferric iron (Fe^3+^), from ferrous iron (Fe^2+^). To perpetuate the cycle, Fe^3+^ is then reduced back to Fe^2+^ by the oxide ion (O^2−^). NO serves as a potent antagonist to ROS chemistry at multiple stages, scavenging ferrous iron, high-valent iron (iron oxo), and H_2_O_2_ [16] (Figure 1).

Various sources of RNS play pivotal roles in cell signaling and contribute to inducing cellular and physiological stress [17]. The metabolism of NO by cells follows first order, while the autoxidation of NO is second order. The autoxidation of NO is significant not only in vitro but also under specific in vivo conditions and serves as a crucial source of RNS. The second-order dependence of NO indicates that the reaction’s half-life changes in proportion to NO concentration. Consequently, at low NO concentrations, the half-life concerning this autoxidation reaction can extend from minutes to hours, allowing for the occurrence of alternate reactions. However, with increasing NO levels, the reaction rate escalates exponentially, particularly in concentrated areas with high iNOS activity [11]. The gasses, NO and O_2_, exhibit 8–10 times higher concentrations in membranes, rendering the NO/O_2_ reaction 100 times faster than in hydrophilic regions [18]. Thiols, such as those associated with RAS, EGFR, and Src, can undergo nitrosation in these conditions, activating numerous oncogenic pathways. Generally, the rate of reaction with di-nitrogen trioxide N_2_O_3_ is significantly higher, nearly reaching diffusion-controlled levels, especially in low pKa thiols like GAPDH [11].

An additional source of NO and RNS comes from nitrite, either under acidic conditions or through oxidation by ROS interactions. Within acidic organelles like lysosomes and phagosomes, nitrite can undergo conversion, initially forming N_2_O_3_, which can further dissociate into NO and NO_2_. This equilibrium establishes a source from which NO can diffuse in an O_2_-dependent manner, interacting with different molecules [19] (Figure 2A; Figure 3). Furthermore, within lysosomes, this equilibrium facilitates thiol nitrosation and tyrosine nitration [20]. This RNS reservoir plays a crucial role in controlling viral replication, particularly within lysosomes. Nitrosation of proteases and other targets inhibits viral activity [21]. The density of iNOS-expressing cells determines the NO level and its diffusion from a source. As shown in Figure 2B, higher densities of iNOS-expressing cells coincide with higher levels of NO and increased diffusion distance from the source.

A variety of reactions involving the interaction of NO and nitrite with metal peroxides play a significant role in biological processes. In hypoxic conditions, reduced metal complexes facilitate nitrite reduction to NO, inducing vasodilation. Conversely, the oxidation of nitrite, catalyzed by ROS-generating reactions, leads to the formation of NO_2_. Nitrite undergoes oxidation to NO_2_ via the Fenton/peroxidase mechanism, leading to nitration [22,23] (Figure 1). In summary, NO and nitrite engage in specific reactions with distinct targets, aligning with temporal and spatial orientations.

## 3. Understanding the Dynamics of NO Diffusion and Consumption for the Generation of High and Low NO Concentrations 

Similarly to oxygen, NO is a gas with diffusible properties that exhibits higher solubility in hydrophobic compartments than in aqueous environments [24]. Constitutive endothelial nitric oxide synthase (eNOS) generates NO at lower fluxes, where NO diffuses from the site and reacts with HbO_2_ in nearby vessels, representing a predominant consumptive mechanism [25]. Conversely, inducible nitric oxide synthase (iNOS), encoded by the NOS2 gene, produces NO at higher concentrations, allowing sustained NO levels. At these concentrations, NO diffuses and is also consumed by cells in an O_2_-dependent manner (Figure 2A; Figure 3).

Interestingly, the rate of production/consumption of NO is similar in normoxic and hypoxic regions. This ultimately results in comparable NO fluxes in both areas, as both NO synthesis and consumption depend on oxygen concentration. Thus, at high O_2_ concentrations, NO synthesis and metabolism are maximized. On the other hand, at low O_2_ concentrations, both NO production and metabolism are reduced. Besides regulating NO production/consumption, O_2_ also impacts NO half-life and thus its diffusion rate. As mentioned, high levels of O_2_ increase NO metabolism reducing its half-life and consequently decreasing the distance of diffusion from the source. The opposite is true under low levels of O_2_ (Figure 2A). This is important because as detailed in the next topic of this review, the biological effects of NO are temporal-, spatial- and concentration-dependent. Consequently, the combination of production, consumption, and diffusion rates plays a crucial role in establishing a particular NO flux. Thus, the diffusion of NO from its source can be envisioned as establishing concentration topography, influencing cells based on their distance to the source [2].

## 4. Cellular Signaling and NO Levels

Levels of NO impact the physiology and pathology of biological processes in a concentration-dependent manner [11,12]. Constitutive isoforms like eNOS and neuronal nitric oxide synthase (nNOS) generate lower NO levels to maintain circulatory and neuronal systems homeostasis, respectively [26]. On the other hand, the inducible isoform, iNOS, can produce higher levels, leading to noxious effects with deleterious consequences (Figure 3). Dysregulated cytokine production, including tumor necrosis factor-alpha (TNF-α), interferon-gamma (INF-y), interleukin-1 alpha (IL-1α), and interleukin-6 (IL-6), can induce high levels of iNOS via nuclear factor-kappa B (NF-kB). This results in increased NO synthesis [27,28] and this dysregulation is observed in various chronic diseases like inflammatory bowel diseases and cancer [8,29].

The concentration-dependent signaling of NO and its RNS elucidates numerous distinctive processes in cancer and other diseases. Prior studies utilizing NO donors, such as diazeniumdiolates or NONOates, have proved to be an essential tool for mapping NO concentration and its effects on precise cellular signaling within physiologic and pathologic systems [30,31]. In vitro models employing various concentration temporal profiles have identified three distinct cellular signaling levels associated with cancer-related mechanisms. The first involves cGMP-related levels (<100 nM), the second encompasses oncogenic signaling levels (100–500 nM), and the third signifies nitrosative stress signaling (>500 nM) [3].

As illustrated in Figure 3, NO levels influence diverse mechanisms driving various oncogenic properties. In liver, prostate, pancreatic, and breast cancer, NO mechanisms have been identified at cGMP-related levels [32] These mechanisms involve a cGMP pathway that triggers ERK phosphorylation, subsequently activating TACE (ADAM17) to elevate NOTCH and EGF. cGMP pathway also influences ADAM10, which plays an important role in extracellular matrix (ECM) remodeling [33]. VEGF is also activated in these levels (<100 nM) promoting angiogenesis. At the higher level of nitrosative signaling (100–500 nM), NO interaction with membrane-bound thiols of low pK_a_ and non-heme iron, such as prolyl hydroxylase, leads to an increase in major oncogenic signaling pathways: such as HIF-1, EGFR/Akt, RAS/ERK, TGF-β, Nrf2, and Aryl hydrocarbon receptor (AhR). At level 3 (>500 nM), the primary focus is the induction of p53 leading to cell cycle arrest and ultimately cell death. Under normal circumstances, p53 inhibits NOS2 by preventing its expression, leading to apoptosis of NOS2-expressing cells [2,34]. However, many cancers have mutated p53, losing the ability to inhibit NOS2 expression and other protumorigenic pathways [35], and chronic exposure to NO increases p53 mutations [36].

## 5. NO in Different Cancer Types

In the context of cancer, NO plays a crucial role in different aspects of cancer progression and response to treatment. This free radical is implicated in driving three key factors associated with unfavorable clinical outcomes in cancer: metastasis, resistance to chemo and radiotherapy, and immune suppression [8,37,38]. While some studies suggest positive effects of NO [39,40,41], the majority link high NOS2 expression to poor survival across a spectrum of hematological and solid tumors, including breast [8], pancreatic [42], nasopharyngeal carcinoma [43], colorectal [44,45], gastric [46], esophageal [47], prostate [48], melanoma [49], Hodgkin’s lymphoma [50], ovarian [51], squamous cell carcinoma [52], renal cell carcinoma [53], glioma [54], chondrosarcomas [55], Barrett’s esophageal adenocarcinoma [56], and in HCV-infected liver cancer patients [57].

Data extracted from the TCGA indicates a significant upregulation of NOS2 in ER- breast cancer patients compared to normal tissue, with a fold change of 13.9. Conversely, COX2 expression was decreased in the same patient cohort, although with a fold change of only 0.7 (Figure 4). These findings corroborate our earlier study, demonstrating the dynamic variation in NOS2 expression across estrogen receptor-negative ER- breast tumor tissues, contrasting with the relatively stable expression of COX2 [8]. Furthermore, Prueitt and colleagues analyzed NOS2 expression in a cohort of 248 breast cancer patients and reported that 29% and 41% of the patients exhibited moderate and high NOS2 expression, respectively [58].

For instance, Stage II colorectal cancer patients with NOS2-high tumors experienced significantly reduced disease-free survival [59,60]. Similarly, elevated NOS2 tumor expression predicted poor survival in Stage III melanoma patients [61]. These findings underscore the association of NOS2 with higher tumor grade, increased lymph node positive status, and enhanced tumor vascularity, suggesting NOS2’s role as a promoter of cancer progression [59,60,61]. Mechanistically, NO is implicated in increased proliferation, adhesion, and migration of tumor cells [62]. NO has also been shown to promote angiogenesis and regulate the activity of a variety of matrix proteins associated with cancer progression and drug resistance [63].

While NOS2 can be considered as a univariate predictor of outcomes, its expression is not isolated from interactions with other significant inflammatory mechanisms. COX-2 is a pro-inflammatory enzyme that is often linked with NOS2 expression in inflammatory conditions and has also been correlated with adverse outcomes in cancer [7]. This mutual relationship between NOS2 and COX-2 has been identified as a mediator of cancer progression in different cancer types such as head and neck squamous cell carcinoma [64], breast [8], colon [44,65], and non-small cell lung cancers [66]. In addition, recent studies from our research group have revealed connections between elevated tumor NOS2 and COX-2 co-expression, chronic inflammation, and the development of more aggressive tumor phenotypes in ER^-^ breast cancer [8]. Similarly, sustained co-expression of these pro-inflammatory markers promotes tumor progression in breast [7] and gastric cancers [67], as well as glioblastoma [68]. In summary, inflammation in the tumor microenvironment (TME) and NOS2 and COX-2 co-expression play pivotal roles in the mechanisms driving unfavorable clinical outcomes.

The interaction between NOS2 and COX-2 within the TME establishes a positive feed-forward loop, each activating pathways that enhance the expression of the other. The resulting products, NO and PGE_2_, create a multidimensional feedforward loop involving Th1 cytokines such as IL-6, IL-1, TNF-⍺, and IL-8, driving different mechanisms that contribute to metastasis, chemoresistance, and the development of cancer stemness.

In the TME, NO and PGE_2_ can induce immunosuppressive factors like IL-10 and TGF-β (Figure 5). NOS2 and COX2 expression have prognostic values in ER- breast cancer with hazard ratios (HR) of 6 [7] and 2.72 [69], respectively. Additionally, the expression of both markers is associated with a remarkable HR of 21 in five years. The synergistic action of these two inflammatory enzymes serves as a mechanism influencing almost every oncogenic pathway and immunosuppressive mechanism, ultimately leading to unfavorable clinical outcomes [7].

In silico analysis corroborates the prognostic value of NOS2 and COX2 in TNBC patients [70,71]. Both markers are highly expressed in TNBC, and their expression is correlated with poor survival outcomes by promoting cancer stemness. Additionally, the co-expression of both markers was correlated with immunosuppression and metastasis in ER- breast cancer patients [8,70].

Under normal conditions, NOS2 and COX2 contribute to establishing a Th1 (IFN-γ) environment. During wound healing and tissue restoration, Th1 response plays a crucial role in restoring tissue homeostasis. In macrophages, an increase in NO leads to elevated p53 activation, inducing apoptosis and subsequently reducing the M1 macrophage population [72,73]. However, in cancer, NOS2 and COX2 can persist for extended periods, activating ATR/ATM, and leading to p53 stabilization [36]. P53, a major negative regulator of both NOS2 and COX2, binds to the TATA box, inhibiting the expression of these enzymes [74]. In p53-competent immune cells and other host cells, a high NO flux would cause a decrease in NOS2 and COX2. Interestingly, high NOS2 expression is associated with increased p53 mutation in breast cancer patients [30]. Additionally, the examination of NOS2 and p53 competency (null or mutant) is linked to elevated NOS2, supporting the negative relationship between NOS2 and p53 [75]. Because p53 is a negative regulator of NOS2 and COX2 expression, these results suggest that increased p53 mutation in tumor cells promotes persistent tumor NOS2 and COX2 expression.

## 6. Induction of NOS2 and COX2 in Cancer Cells

Over the past five decades, the activation of NOS2 and COX2 in murine macrophage models has been demonstrated to occur through the combination of IFN-γ with either cytokines (IL-1β and TNF-⍺) or lipopolysaccharide (LPS). While the most significant induction in murine myeloid cells was observed with a combination treatment of IFN-γ and LPS, the combination of INF-γ with TNF-⍺ and IL-1β yielded considerably lower effects [76]. On the other hand, the optimal induction of NOS2 and COX2 in murine and human tumor cells requires IFNγ, and its combination with TNF-⍺ and IL-1β was more effective than with LPS. These observations suggest variations in the pathways leading to NOS2 induction in different cell types. Comparing murine macrophages and tumor cells, a temporal distinction emerges, where the maximum NOS2 in macrophages and murine tumor cells occurs within 8 h to 24 h [77]. In human tumor cells, however, expression is delayed, occurring between 24 h to 48 h [8] (Figure 6). This temporal disparity implies that NOS2 expression in different cells may exhibit varying levels and sustainability of NO based upon orientation and timing of NOS2 expression.

A crucial aspect of NOS2 and COX2 is their orthogonal cellular expression in distinct cells. Traditionally, the Western blot concept assumed a homogeneous expression of both NOS2 and COX2 in every cell. However, microscopic examination and single cell-RNA-sequencing (sc-RNA-seq) have revealed orthogonal expression in both human and murine cells, with some cells expressing NOS2 and others expressing COX2 [8,77]. These results suggest that uniform stimulation of IFN-γ with other cytokines can create distinct phenotypes of cells expressing NOS2 or COX2. The orthogonal expression of NOS2 and COX2 implies unique roles for these cells in the TME, pointing to novel cellular neighborhoods. These cellular niches collaborate to sustain the expression of NOS2 and COX2 while also driving various oncogenic and immunosuppressive mechanisms [2,3] (Figure 5). Therefore, the spatial orientation of NOS2 and COX2 in the tissue provides crucial insights into the cellular neighborhoods that contribute to poor outcomes.

## 7. Human Breast Cancer and the Development of NOS2/COX2 Metastatic Niche Require CD8/IFNγ

The requirement of IFN-γ and TNF-⍺/IL-1β for the upregulation of NOS2 and COX2 in human tumor cells implies that the microenvironment inducing NOS2-rich regions requires cells present that are releasing IFN-γ and other proinflammatory cytokines. Lymphoid cells, including CD8^+^ T effector cells, CD4^+^ Th1, and NK cells, are potential sources, serving as major cytokine contributors under inflammatory conditions [78]. In our previous study [8,76], we demonstrated that stromal-restricted CD8^+^ T cells and limited lymphoid aggregates were closely positioned but orthogonal to tumor regions exhibiting high levels of NOS2 and COX2. Additionally, IFN-γ staining corresponded to regions with elevated NOS2 levels. On the other hand, in “immune-cold” areas of the tumor with low CD8^+^ T cells (immune deserts), there was notably less NOS2 and IFN-γ. Interestingly, NOS2 and COX2 were not co-expressed in the same cell, and areas with higher concentrations of NOS2 were associated with a specific metastatic cellular landscape. NOS2 was identified at the tumor-stromal interface, while COX2 was present in the vicinity of immune desert regions closer to the tumor core. This NOS2/COX2 spatial configuration in specified areas of the tumor indicated an increased metastatic potential. In summary, this spatial arrangement suggests a significant association between NOS2 and COX2 niches that drive poor clinical outcomes in ER- breast cancer (Figure 5).

Although NOS2/COX2 regions do not span the entire tumor and are not linked to a broader immune desert, these small focal points suggest a mechanism of cancer progression associated with poor outcomes. This association of NOS2 and COX2 induction and cancer progression was also observed in in vitro models using NO donors [7]. The crucial factor in the development of the NOS2 niche is the stroma restriction of CD8^+^ T cells [8]. Extrapolating from murine models of triple-negative breast (TNBC), it has been demonstrated that tumor NOS2 and COX2 play a vital role in CD8^+^ T cell stroma restriction, thus preventing their infiltration into the tumor core [76]. Therefore, IFN-γ originating from a proximal source, such as stroma-restricted CD8^+^ T effector cells, is pivotal in the formation of aggressive tumor NOS2 and COX2 niches that drive the NO and PGE_2_ feedforward loops, ultimately leading to increased metastasis and cancer stem cell (CSC) characteristics [7,8]. The identification of these niches presents novel therapeutic opportunities for the treatment of patients with these tumor signatures using clinically available NOS/COX inhibitors. Also, the discovery of inhibitory agents that limit the development of NOS2/COX2 niches could improve clinical outcomes by reducing metastasis, chemoresistance, and immunosuppression. This offers a potential playbook for screening novel and clinically available agents for the treatment and prevention of more aggressive cancers, such as TNBC.

## 8. Alternative Therapeutic Approaches for Preventing NOS2/COX2 Cellular Niche Development 

A variety of direct and indirect strategies can be employed to diminish the formation of NOS2 and COX2 niches. There are different non-steroidal anti-inflammatory drugs (NSAIDs) that can target iNOS and/or COX2 and exhibit potential repurposing capacity for treating different types of cancers (Table 1). Targeting NOS2 and COX2 can involve: (1) reducing expression or targeting critical downstream effectors (indirect approach), and (2) inhibiting NO and PGE_2_ production from NOS2 and COX2 (direct approach) (Figure 7). These strategies are applicable in different contexts including chemoprevention and co-adjuvant treatment aiming to prevent cancer development, metastasis, and tumor recurrence. In preventive approaches, the focus is on well-tolerated long-term therapies with minimal side effects, while in contrast, more aggressive measures can be considered during active treatment phases [79].

### 8.1. Direct Methods of NOS2 and COX2 Inhibition

Direct methods involve the use of compounds that decrease NO or PGE_2_ levels by inhibiting iNOS (NOS2) and COX2 enzyme activity, respectively (Figure 7). NOS and COX enzymes have both constitutive and induced isoforms: constitutive (nNOS, eNOS, and COX1) and induced (iNOS and COX2). Pan-inhibitors for both NOS and COX, which target both constitutive and induced isoforms, are available [154]. The advantage of pan-inhibitors lies in their ability to address various cancer-related mechanisms involving eNOS, such as angiogenesis and immune response regulation [155]. COX1 in certain situations inhibits apoptosis and processes other prostaglandins (PGs). Selective iNOS inhibitors, such as acetamidine compounds, were administered to patients for over two years with minimal side effects, offering an alternative to pan-inhibitors [156]. Alternatively, aminoguanidine (AG) is an iNOS-specific inhibitor utilized in humans for conditions like diabetes [157]. In murine models, AG significantly reduced metastasis [83]. However, AG is a weaker iNOS inhibitor compared to acetamidine compounds, such as 1400 W, and has off-target effects on other enzymes, including inhibiting diamine oxidase involved in polyamine synthesis [158].

There are FDA-approved compounds targeting COX2 that are clinically available. Celecoxib compounds and other COX2-specific agents provide targeted inhibition of PGE_2_ [159]. The use of these COX2-specific compounds presents a potential avenue for treatment. An alternative approach to PGE_2_ inhibition involves increasing 15-PDGH, an enzyme that metabolizes PGE_2_. NSAIDs, such as diclofenac and indomethacin, can inhibit COX while simultaneously boosting 15-PGDH. Some compounds have been developed to enhance 15-PDGH activity as an alternative to COX inhibition, however, these drugs can lead to gastrointestinal and cardiovascular effects [160]. Concerns about cardiovascular side effects with long-term use of pan NOS and COX inhibitors have been raised. However, in a recent phase 1/2 clinical trial, amlodipine and low-dose aspirin were given to prevent hypertension and thromboembolism, respectively in chemoresistant TNBC patients receiving the NOS inhibitor L-NMMA combined with docetaxel [161].

Moreover, in the same phase 1/2 clinical trial by Chung et al. [161], patients with chemorefractory breast cancer, locally advanced breast cancer (LABC), or metastatic TNBC were administered a regime composed of L-NMMA and docetaxel. It was reported a 45.8% overall response rate, 81.4% for patients with LABC, and 15.4% for metastatic TNBC patients. Additionally, another phase 1b/2 clinical trial with 15 metaplastic TNBC patients, a more aggressive form of TNBC, showed that the combination of L-NMMA and docetaxel results in an overall response of 20%, and progression-free survival (PFS) and overall survival (OS) of 4.5 months (range 3–7 m) and 12.8 months, respectively. Thus, inhibition of the nitric oxide pathway represents a promising and novel therapeutic option for TNBC. Further investigation with substantial cohorts is required to better understand the clinical benefits of NOS inhibitors in association with conventional chemotherapy [162].

### 8.2. Indirect Methods

An alternative strategy for direct inhibition involves targeting the induction of NOS2 and COX2, as well as the downstream products in their cellular signaling pathways. A variety of anti-inflammatory agents that target IFN-γ signaling are available [163,164]. However, a challenge arises due to the dichotomous nature of IFN-γ: while it can induce NOS2, it is also essential for a successful antitumor response [165]. Similar concerns arise when targeting NFκ-B and other components of the NOS2 and COX2 cellular signaling pathways, as this may interfere with critical anti-tumor pathways [166] (Figure 7). Therefore, the optimal strategy depends on the timing and context to ensure the most effective approach.

Different natural compounds aim to target the expression of NOS2 and COX2 [167]. These nutraceuticals encompass simple compounds like thiol-based agents and agonists of the antitumor PP2A [168]. Some dietary-based compounds like sulforaphane, have been transformed into pharmacological drugs that may have a role in this context [169]. Some of these nutraceutical compounds function as phase 1 enzyme inhibitors and phase 2 activators (Figure 7). Phase 1 enzymes are associated with procarcinogens, while phase 2 enzymes are part of anticarcinogenic systems [170].

Phase 1 enzymes exert various effects beyond inducing pro-carcinogenic P450 enzymes. They also play a role in regulating DNA and histone methylation in enzymes like BRCA1 [171]. In addition, these enzymes can induce AhR activation promoting COX2 expression [172] (Figure 7). The products of indoleamine 2,3-dioxygenase (IDO), induced by IFN-γ, can enhance AhR, leading to the production of kynurenines that exert diverse impacts on immunosuppression [173]. AhR activation is further linked to the induction of epithelial-mesenchymal transition (EMT), metastasis, and cancer stemness [174]. Hence, conditions that activate AhR, leading to increased COX2 and NOS2 via S-nitrosation enhance cellular resistance to treatment and promote EMT, metastasis, and immunosuppression.

In contrast, phase II enzymes are activated through NrF2/keap1, initiating a cascade that primarily protects against xenobiotics [175]. NrF2 increases antioxidant compounds, such as HO-1, playing a role in chemoprevention [176]. However, when these antioxidant defenses are activated in the context of cancer it leads to chemoresistance and immune suppression [177]. NO can activate NrF2 through S-nitrosation of keap1, releasing NrF2 and rendering cells more resistant to xenobiotics and electrophilic stress [178].

As mentioned earlier, several compounds classified as phase 1 inhibitors exhibit the ability to activate phase 2 enzymes. A subset of sulfur-based compounds, mainly found in cruciferous vegetables, exemplifies this dual action, including dithiolethiones [179] and isothiocyanates [180]. These thiol-based compounds extend their impact beyond phase transitions, demonstrating potent effects on additional targets, including the inhibition of NFκB [181,182]. Among dithiolethione compounds, anethole dithiolethione (ADT) stands out for its ability to activate PP2A by targeting SET, a pro-oncogenic protein that inhibits various anti-tumor proteins such as NMEM23 (known for its anti-metastatic properties) and PP2A [183,184,185]. PP2A is a crucial serine-threonine phosphatase that serves to inhibit numerous oncogenic pathways activated by NO (Figure 7). From a cancer cell-centric perspective, the activation of PP2A and the inhibition of pathways like NFκB, Akt, and ERK could be indicative of tumor regression [186,187,188]. These nutraceuticals present an opportunity to explore insights gained from spatial analysis of NOS2/COX2 and the chemical biology of NO and PGs for potential therapeutic applications in cancer.

## 9. Conclusions

Understanding the localization of NOS2/COX2 in tumors offers valuable insights and an opportunity to devise new approaches for targeting and preventing the progression of advanced cancers. The discussion above emphasizes that small, concentrated regions of NOS2-expressing tumor cells, compared to the entire tumor, can significantly impact survival. These small NOS2-rich areas generate elevated levels of NO, primarily formed through IFN-γ/cytokine stimulation. Identifying these NOS2 niches within tumors highlights a specific cellular mechanism associated with adverse outcomes. In addition, the pro-inflammatory enzyme, COX2, can potentiate NOS2 expression through a positive feed-forward loop. Therefore, spatial implications of NOS2/COX2 and the chemical biology of NO and RNS serve as a crucial roadmap for the design of new clinical trials supporting the development of innovative therapies that disrupt the nitric oxide signaling pathway and ultimately reduce immunosuppression, cancer stemness, and metastasis.

## Figures and Tables

**Figure 1 ijms-25-06103-f001:**
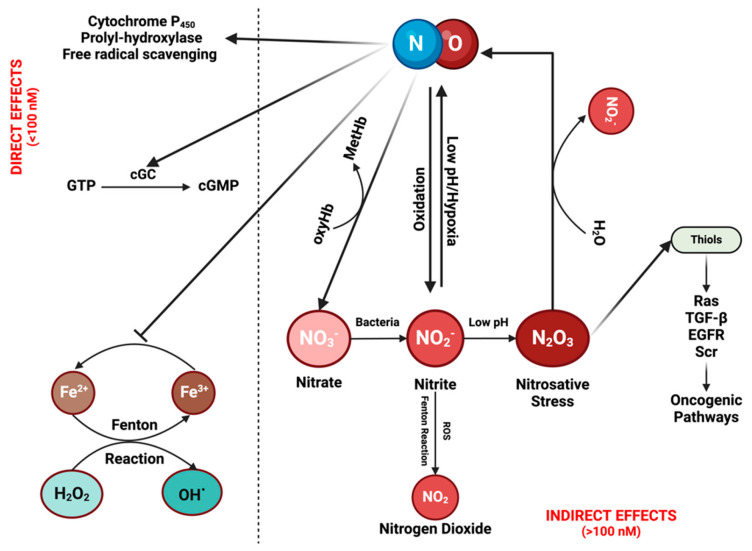
**The chemical biology of nitric oxide (NO) and reactive nitrogen species (RNS):** NO can exert its effects either directly or indirectly. Direct effects are associated with low NO levels (<100 nM). At these concentrations, NO can activate the enzyme cyclic guanylyl cyclase (cGC), initiating the conversion of guanosine triphosphate (GTP) into cGMP. cGMP then serves as a signaling molecule in various physiological processes, such as vasodilation and neurotransmission. Moreover, it contributes to drug detoxification by interacting with proteins from the cytochrome P450 family, which can also eliminate NO itself. In addition, NO can induce the hypoxia-inducible factor (HIF) pathway through the inhibition of prolyl-hydroxylase, while also functioning as a free radical scavenger by inhibiting the Fenton reaction and reacting with reactive oxygen species (ROS). Indirect effects of NO are driven by its reactive nitrogen species (RNSs). These reactive species can undergo various reactions, leading to their recycling back into NO while simultaneously activating diverse oncogenic pathways. Figure Made in BioRender.com (accessed on 21 February 2024).

**Figure 2 ijms-25-06103-f002:**
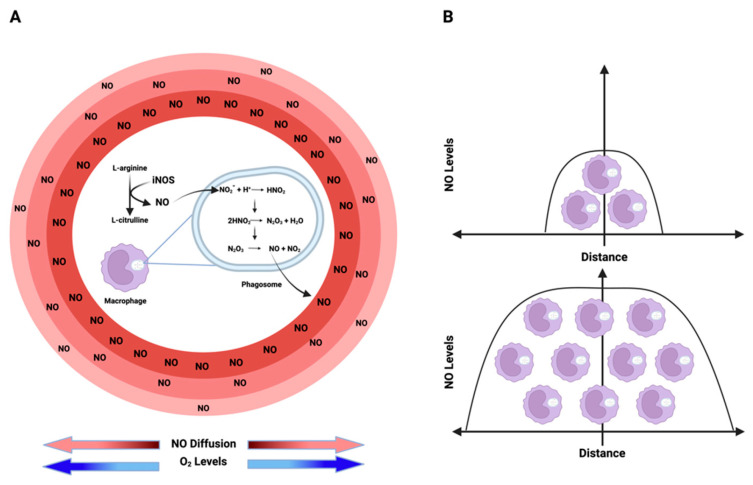
**Nitric oxide (NO) kinetics:** (**A**) NO is enzymatically synthesized through the oxidation of L-arginine by iNOS. Conversely, nitrite (NO_2_^−^), the product of NO, acts as an alternative NO reservoir in acidic environments, such as phagosome/lysosome lumens. In these regions, nitrite undergoes a series of reactions, ultimately generating NO and NO_2_, which then diffuse from the source to the surroundings in an O_2_-dependent manner due to O_2’_s impact on NO half-life creating an NO flux gradient with higher levels around the source and lower levels as the distance from the source increases. (**B**) The extent to which NO can diffuse from its origin and the levels of this free radical in a specific environment depend on the cell density of iNOS-expressing cells. Higher densities of NO-producing cells result in elevated NO levels and an increased diffusion distance. Figure Made in BioRender.com (accessed on 21 February 2024).

**Figure 3 ijms-25-06103-f003:**
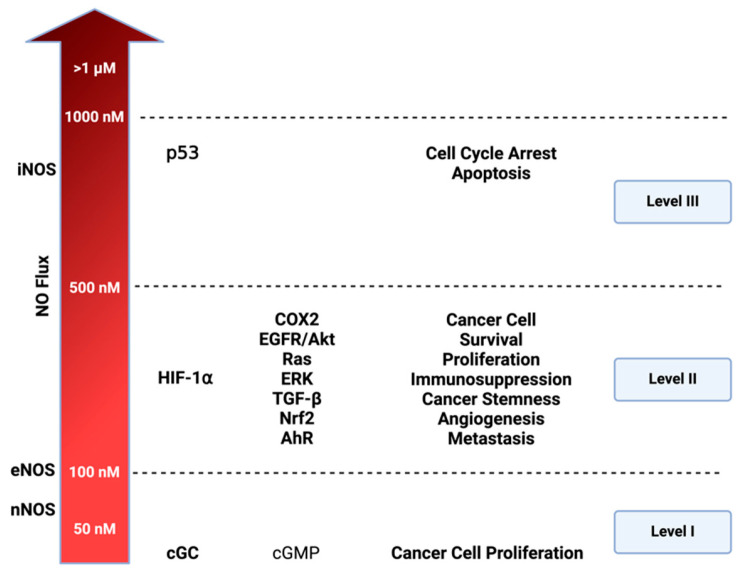
**Cellular signaling and nitric oxide (NO) levels:** NO effects vary based on its concentration. Three distinct nitrosative levels are identified: Level 1 (<100 nM) is linked to cGMP-related cell signaling, contributing to cancer cell proliferation; Level 2 (100–500 nM) involves HIF-1⍺ activation and subsequent pathways promoting cancer progression; Level 3 (>500 nM) is associated with toxic NO effects, primarily triggered by p53 activation, leading to cell cycle arrest and apoptosis. Figure Made in BioRender.com (accessed on 21 February 2024).

**Figure 4 ijms-25-06103-f004:**
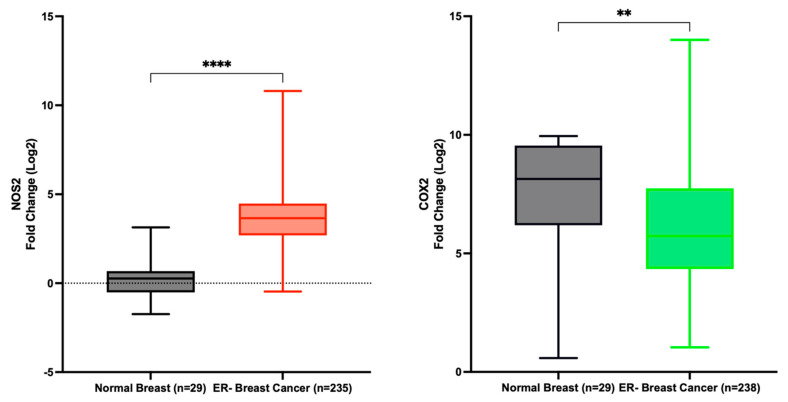
NOS2 and PTGS2 (COX2) expression in normal breast and ER- breast cancer. **** *p* < 0.0001; ** *p* = 0.0032.

**Figure 5 ijms-25-06103-f005:**
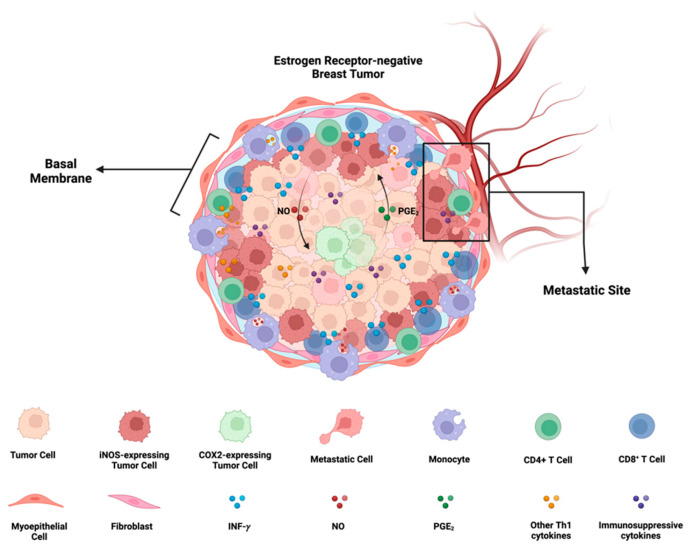
**iNOS/COX2 co-expression and its effects on ER^-^ breast cancer progression:** In ER- breast cancer, iNOS and COX2 exhibit orthogonal expression patterns. iNOS niches are located at the stroma-tumor interface, preventing the penetration of CD8^+^ T cells into the tumor, and are associated with a metastatic phenotype. On the other hand, COX2 niches are deeper within the tumor core, near immune-desert regions. There exists a positive feedforward loop between iNOS and COX2, where iNOS-derived nitric oxide (NO) induces COX2 activity, and COX2-derived PGE_2_ induces iNOS. The formation of these iNOS and COX2 niches depends on the presence of INF-γ-secreting cells, such as CD8+ T cells, and may be potentiated by other Th1 cytokines like TNF-⍺ and IL-1β. Furthermore, NO and PGE_2_ have the potential to induce the production of immunosuppressive cytokines such as IL-10 and TGF-β. Figure Made in BioRender.com (accessed on 24 February 2024).

**Figure 6 ijms-25-06103-f006:**
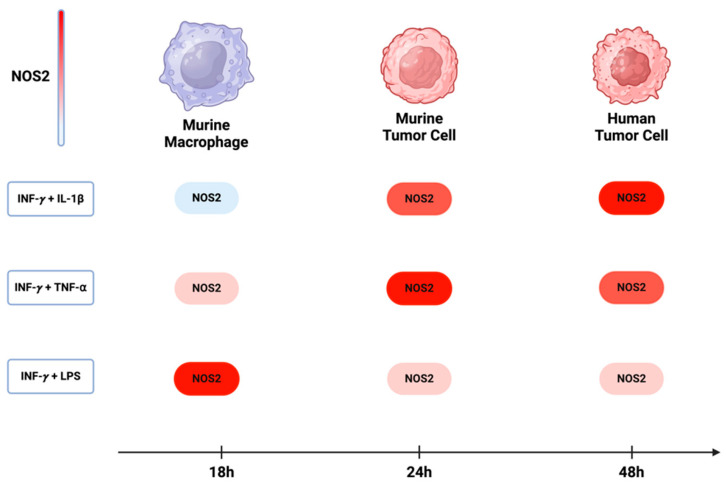
**NOS2 induction in murine and human cells:** In murine macrophages, the maximum induction of NOS2 occurs with a combination of INF-y and LPS. However, in both murine and human tumor cells, the highest expression of NOS2 is achieved when INF-y is combined with either TNF-⍺ or IL-1β. The induction of NOS2 in murine macrophages occurs approximately 18 h after treatment, while in murine and human tumor cells, there is a delay, with induction occurring around 24 h to 48 h. Figure Made in BioRender.com (accessed on 24 February 2024).

**Figure 7 ijms-25-06103-f007:**
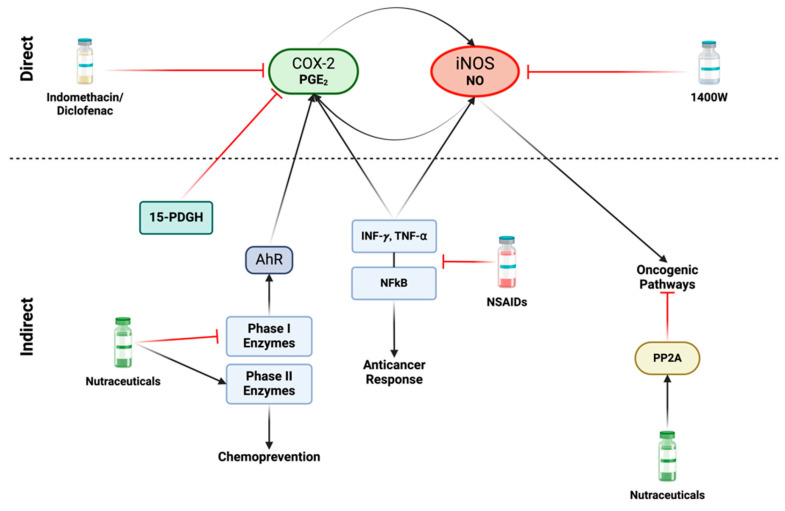
**Direct and indirect approaches for inhibiting iNOS and COX2:** Various nonsteroidal anti-inflammatory drugs (NSAIDs), including indomethacin and diclofenac, as well as Celecoxib can directly inhibit COX2. Specific iNOS inhibitors, such as acetamidine compounds like 1400 W, demonstrate highly efficient and targeted effects. In terms of indirect approaches, the focus shifts to factors that induce iNOS and/or COX2 activity. A variety of NSAIDs play a role in inhibiting the production and release of proinflammatory cytokines, such as INF-γ and TNF-α, leading to subsequent inhibition of the NFκB pathway. However, inhibition of these proinflammatory mediators poses a challenge, as they are also crucial to anticancer responses, and their suppression may promote cancer progression. An alternative strategy to disrupt the iNOS/COX2 feedforward loop involves nutraceuticals, such as isothiocyanates and dithiolethione compounds. These compounds exhibit the capability to inhibit carcinogenic phase I enzymes while simultaneously inducing phase II detoxification enzymes, thereby promoting chemoprevention. Phase I enzymes can activate the Aryl hydrocarbon receptor (AhR), leading to COX2 activity induction. These natural compounds also serve as activators of the tumor suppressor protein phosphatase 2A (PP2A). This enzyme acts as an antagonist to various oncogenic pathways activated by nitric oxide. Figure Made in BioRender.com (accessed on 26 February 2024).

**Table 1 ijms-25-06103-t001:** List of non-cancer drugs that have shown some evidence of anticancer activity. Data extracted from https://www.anticancerfund.org/database-repurposing-drugs-oncology (accessed on 14 May 2024) and https://pubmed.ncbi.nlm.nih.gov/ (accessed on 14 May 2024).

**Drug:**	Acetylsalicylic Acid
**Synonym:**	Aspirin
**Indication:**	Analgesia, swelling, prophylaxis of venous embolism and further heart attacks or strokes.
**Targets:**	PTGS1; **PTGS2**; AKR1C1; PRKAA1; EDNRA; TP53; HSPA5; RPS6KA3; NFKBIA; TNFAIP6; CASP1; CASP3; IKBKB; MAPK1; CCND1; MYC; PCNA; NEU1.
**Cancer:**	Melanoma, colorectal cancer, colon, rectum, esophagus, stomach, liver, pancreas, small intestine, head and neck, brain tumors, meningioma, melanoma, thyroid, non-Hodgkin lymphoma, and leukemia.
**Study:**	[80,81,82]
**Drug:**	Aminoguanidine
**Synonym:**	Pimagedine
**Indication:**	Diabetic retinopathy, insulin-resistant disorder, rheumatoid arthritis.
**Targets:**	CCR2, **NOS2**.
**Cancer:**	Pancreatic cancer, breast cancer, prostate cancer, hepatocarcinoma.
**Study:**	[83,84,85]
**Drug:**	Diclofenac
**Synonym:**	
**Indication:**	Osteoarthritis, rheumatoid arthritis, ankylosing spondylitis.
**Targets:**	**PTGS2**; PTGS1.
**Cancer:**	Fibrosarcoma, colorectal cancer, neuroblastoma, ovarian cancer, pancreatic cancer.
**Study:**	[83,86,87,88,89,90,91]
**Drug:**	Cannabidiol
**Synonym:**	
**Indication:**	Seizures associated with Lennox-Gastaut & Dravet syndromes.
**Targets:**	CNR1; CNR2; GPR12; GLRA1; GLRA1; GLRA3; GPR18; GPR55; HTR1A; HTR2A; CHRNA7; OPRD1; OPRM1; PPARG; TRPV1; CACNA1G; CACNA1H; CACNA1I; TRPA1; TRPM8; TRPV2; TRPV3; TRPV4; VDAC1; HTR3A; ADORA1; PTGS1; **PTGS2**; ACAT1; CYP17A1; HMGCR; GSR; GPX1; IDO1; CYP1B1; NQO1; CAT; CYP3A5; CYP2D6; SOD1; CYP1A2; CYP3A7; AANAT; NAAA.
**Cancer:**	Lung cancer, glioma, prostate, colorectal cancer, breast cancer, uterine cancer.
**Study:**	[92,93,94,95]
**Drug:**	Celecoxib
**Synonym:**	
**Indication:**	Osteoarthritis, rheumatoid arthritis, ankylosing spondylitis, juvenile rheumatoid arthritis, acute pain, primary dysmenorrhea.
**Targets:**	**PTGS2**; CA2; CA3; CDH11; PDPK1; NEU1.
**Cancer:**	Osteosarcoma, oral squamous cell carcinoma.
**Study:**	[96,97]
**Drug:**	Diflunisal
**Synonym:**	Osteoarthritis, rheumatoid arthritis, mild to moderate pain.
**Indication:**	
**Targets:**	**PTGS2**; PTGS1.
**Cancer:**	Leukemia, colon cancer, prostate cancer.
**Study:**	[98,99]
**Drug:**	Etodolac
**Synonym:**	
**Indication:**	Analgesia
**Targets:**	**PTGS2**; PTGS1; RXRA.
**Cancer:**	Prostate cancer, colon cancer, bladder cancer, breast cancer, head and neck squamous cell carcinoma.
**Study:**	[100,101,102,103,104]
**Drug:**	Etoricoxib
**Synonym:**	
**Indication:**	Osteoarthritis, rheumatoid arthritis, ankylosing spondylitis.
**Targets:**	**PTGS2**.
**Cancer:**	Cervical cancer, colon cancer, breast cancer, pancreatic cancer.
**Study:**	[105,106,107,108]
**Drug:**	Flurbiprofen
**Synonym:**	
**Indication:**	Analgesia
**Targets:**	**PTGS2**; PTGS1.
**Cancer:**	Thyroid cancer, colorectal cancer, non-small cell lung cancer.
**Study:**	[109,110,111]
**Drug:**	Ibuprofen
**Synonym:**	
**Indication:**	Analgesia
**Targets:**	**PTGS2**; PTGS1; BCL2; THBD; FABP2; PPARG; CFTR; PPARA; GP1BA; S100A7.
**Cancer:**	Breast cancer, lung cancer, hepatocellular carcinoma.
**Study:**	[112,113,114]
**Drug:**	Indomethacin
**Synonym:**	Indometacin
**Indication:**	Analgesia
**Targets:**	**PTGS2**; PLA2G2A; PTGS1; PTGR2; PPARG; GLO1; PTGDR2; PPARA; AKR1C3; EIF2AK2; PTGS1.
**Cancer:**	Colorectal cancer, breast cancer.
**Study:**	[76,115,116,117]
**Drug:**	Ketoprofen
**Synonym:**	
**Indication:**	Analgesia
**Targets:**	**PTGS2**; PTGS1; CXCR1.
**Cancer:**	Breast cancer, colon cancer, leukemia, ovarian cancer.
**Study:**	[118,119,120,121]
**Drug:**	Ketorolac
**Synonym:**	
**Indication:**	Post-operative analgesia
**Targets:**	**PTGS2**; PTGS1.
**Cancer:**	Breast cancer, ovarian cancer.
**Study:**	[122,123,124]
**Drug:**	Loxoprofen
**Synonym:**	
**Indication:**	Analgesia
**Targets:**	**PTGS2**; PTGS1.
**Cancer:**	Breast cancer, colon cancer.
**Study:**	[124]
**Drug:**	Meclofenamate
**Synonym:**	Meclofenamic Acid
**Indication:**	Analgesia
**Targets:**	PTGS1; **PTGS2**; ALOX5; KCNQ2; KCNQ3.
**Cancer:**	Non-small cell lung cancer, prostate cancer.
**Study:**	[125,126]
**Drug:**	Mefenamic Acid
**Synonym:**	
**Indication:**	Osteoarthritis, rheumatoid arthritis, analgesia.
**Targets:**	**PTGS2**; PTGS1.
**Cancer:**	Colon cancer, liver cancer, cervical cancer.
**Study:**	[127,128,129,130]
**Drug:**	Meloxicam
**Synonym:**	
**Indication:**	Analgesia
**Targets:**	**PTGS2**; PTGS1.
**Cancer:**	Hepatocellular carcinoma, colorectal cancer, osteosarcoma.
**Study:**	[131,132,133]
**Drug:**	Mesalazine
**Synonym:**	Mesalamine, 5-aminosalicylic acid
**Indication:**	Inflammatory bowel disease
**Targets:**	**PTGS2**; PTGS1; ALOX5; PPARG; CHUK; IKBKB; **NOS2**.
**Cancer:**	Colon cancer, breast cancer.
**Study:**	[134,135]
**Drug:**	Miconazole
**Synonym:**	
**Indication:**	Fungal infections
**Targets:**	ERG11; NOS3; **NOS2**; NR1I2; KCNMA1; KCNA1; KCNJ12; CACNA1C.
**Cancer:**	Glioblastoma, bladder cancer, breast cancer, colon cancer.
**Study:**	[136,137,138,139]
**Drug:**	Minocycline
**Synonym:**	
**Indication:**	Antibiotic
**Targets:**	rpsI; rpsD; IL1B; ALOX5; MMP9; VEGFA; CASP1; CASP3; CYCS; MAPK1; **NOS2**.
**Cancer:**	Colorectal cancer, non-small cell lung cancer.
**Study:**	[140,141]
**Drug:**	Nimesulide
**Synonym:**	
**Indication:**	Analgesia
**Targets:**	**PTGS2**; PLA2G2E; LTF.
**Cancer:**	Pancreatic cancer
**Study:**	[142]
**Drug:**	Sulfasalazine
**Synonym:**	
**Indication:**	Rheumatoid arthritis; ulcerative colitis; active Crohn’s disease.
**Targets:**	ALOX5; **PTGS2**; PTGS1; ACAT1; PLA2G1B; SLC7A11; NFKB1; NFKB2.
**Cancer:**	Gastric cancer, head and neck, glioblastoma, hepatocellular carcinoma, breast cancer.
**Study:**	[143,144,145,146]
**Drug:**	Sulindac
**Synonym:**	
**Indication:**	Relief of signs and symptoms of osteoarthritis, rheumatoid arthritis, ankylosing spondylitis, acute painful shoulder (acute subacromial bursitis/supraspinatus tendinitis), and acute gouty arthritis.
**Targets:**	**PTGS2**; PTGS1; AKR1B1; MAPK3; PPARD; PTGDR2; AKR1B10.
**Cancer:**	Colon cancer, breast cancer, pancreatic cancer.
**Study:**	[147,148,149]
**Drug:**	Tolfenamic Acid
**Synonym:**	Tolfenamate
**Indication:**	Migraine
**Targets:**	PTGS1; **PTGS2**.
**Cancer:**	Pancreatic cancer, esophageal cancer, colon cancer, head and neck cancer.
**Study:**	[150,151,152,153]

## Data Availability

Not applicable.

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
