# Peer review of "NOS2 and COX-2 Co-Expression Promotes Cancer Progression: A Potential Target for Developing Agents to Prevent or Treat Highly Aggressive Breast Cancer"

_ijms, 2024, doi:10.3390/ijms25116103_

Round 1

Reviewer 1 Report

Comments and Suggestions for Authors

The review "NOS2 and COX-2 Co-expression Promotes Cancer Progression: A Potential Target for Developing Agents to Prevent or Treat Highly Aggressive Breast Cancer" is an interesting scientific study with the clinical aspect highlighted by the Authors.                                                                           The review is prepared in a standard way. The figures included in this review are a summary of the issues described.                                                              The most of the references were published within the last 10-15 years, so they are current.                                                                                            Due to the scientific and clinical aspects of the review "NOS2 and COX-2 Co-expression Promotes Cancer Progression: A Potential Target for Developing Agents to Prevent or Treat Highly Aggressive Breast Cancer", I propose that it be accepted for printing in IJMS in its current form.

Author Response

Dear reviewer,

We extend our gratitude for dedicating time to thoroughly review our work. We are pleased to hear that our manuscript met your expectations. Thank you sincerely for your thoughtful assessment.

Best regards,

Leandro Coutinho

Reviewer 2 Report

Comments and Suggestions for Authors

In "NOS2 and COX-2 Co-expression Promotes Cancer Progression: A Potential Target for Developing Agents to Prevent or Treat Highly Aggressive Breast Cancer", Coutinho and colleagues set out to review the literature on a NOS-COX2 co-expression module that might be a targetable node in novel breast cancer therapy.

This is an interesting subject, and the authors do a commendable effort to present an overview of the known literature. I am not an NO or NOS expert, so I will restrict my comments to those I can make as a general cancer scientist with knowledge of target identification and therapy development. 

The review is well-structured, and presents a reasonable case. I miss important data however, that the authors could relatively easily extract from large databases on breast and other cancers in the public domain. The TCGA, R2, and cBioportal websites come to mind, but several others exist. These data could then be combined into clear tables, together with the information from previous literature that is now condensed into a few sentences (e.g. lines 225-231). 

Main issues to be addressed:

1. The authors give as title "NOS2 and COX-2 co-expression" and should therefore address or present large-scale data on the protein (or mRNA) co-expression in breast or other cancers. 

2. The authors should present large-scale data on the correlation between high NOS2 expression and poor survival in the cancer types presented in lines 225-231 of paragraph 5. 

Minor:

Since the authors present NOS-COX-2 as a targetable node, paragraphs 8-10 would benefit from presentation in clear table overviews. In addition, data from large cancer drug databases like e.g. SEER*RX or Redo (https://www.anticancerfund.org/database-cancer-drugs) could be integrated.  

Author Response

Dear reviewer,

We are grateful for the time you invested in meticulously reviewing our work. Your attention to the NOS2/COX2 co-expression is greatly appreciated, as it holds significant importance in understanding the overall survival of ER- breast cancer patients. Our previous research (Basudhar et al., 2017) highlighted this aspect, demonstrating a substantial decrease in overall survival among ER- breast cancer patients co-expressing NOS2 and COX2, compared to those expressing either marker singly or none of these markers. This finding, with a hazard ratio of 21, was based on a cohort study involving 83 ER- breast cancer patients from different hospitals in Maryland. We referenced this work in paragraph 17, lines 8-10. Furthermore, we included bioinformatics studies (paragraph 18) to complement the importance of NOS2 and COX2 on clinical outcomes of ER- breast cancer patients. Regarding the analysis of drug databases, we intend to delve deeper into this matter in future work in which we aim to identify different compounds targeting the NOS2/COX2 feedforward loop, potentially offering therapeutic avenues in the treatment of ER- breast cancer.

Best regards,

Leandro Coutinho

Reviewer 3 Report

Comments and Suggestions for Authors

Dear Editor,

The review article titled NOS2 and SOX2 co-expression promotes cancer progression: A potential target for developing agents to prevent or treat highly aggressive breast cancer by Leandro et al., drafter well.

As we know that breast cancer represents a complex and heterogeneous disease, encompassing various tumor subtypes with distinct clinical behaviors and responses to therapy. Among these subtypes, triple-negative breast cancer (TNBC) stands out for its aggressive nature, high metastatic potential, and limited treatment options.

I would recommend this review article for the publication with minor correction.

Minor changes:

1. Figure 1 should precede Figure 2 as it currently appears at the beginning, followed by Figure 2.

2. Figure 2A is not very clear. All the font size is small. Similarly, Figure 4 is also not clear.

3. Figure 1-6 (except 5) created using BioRender. Author should acknowledge when using software.

4. Author should include about the clinical trials on this study.

5. This review article is lacking on future direction.

Thanks.

Author Response

Dear reviewer,

We extend our gratitude for dedicating time to thoroughly review our work and for your invaluable suggestions, which have significantly enhanced the quality of our manuscript. In response to your recommendations, we have restructured the order of the figures and adjusted the font size accordingly. Additionally, we have revised Figure 2A to enhance clarity. Biorender has been appropriately credited at the end of each figure created using the software. Furthermore, we have provided a more comprehensive overview of the clinical trials referenced, incorporating a recent study to broaden our understanding of the clinical implications of nitric oxide inhibition in breast cancer treatment.

Best regards,

Leandro Coutinho

Round 2

Reviewer 2 Report

Comments and Suggestions for Authors

The author kindly recognize my concerns with this study, but do not sufficiently address them. I therefore have no other choice than to repeat my concerns and reject this manuscript in its current form.

Author Response

Dear reviewer, your comment is highly relevant, which we are currently investigating in detail in separate papers comparing larger databases of specific tumor types.  While informative, the TCGA gene expression database only provides one mechanistic aspect. Nonetheless, we have added a comparison of normal vs tumor gene expression from ER- breast cancer, which shows that NOS2 is elevated in the tumor.  In contrast, COX2 mRNA expression is more ubiquitous in different diseases and as well as cancer, and is more difficult to evaluate its predictive power.   Also, the exact condition of the harvested normal tissue is very important:  for example, if the normal tissue is adjacent to the tumor our data shows extensive COX2 expression throughout both tumor and normal tissues. 

Another key problem with comparing different tumor types is that they are diagnosed and harvested at points relative to disease progression.  When cancers are detected late, they tend to be large immune deserts where NOS2% is low and even COX2 is diluted.  In contrast, breast and colon cancers are harvested at earlier stages in disease progression where elevated tumor NOS2 is associated with inflammatory stroma-restricted lymphoid cells, which we have validated in additional cohorts including DCIS and liver cancer. Examination of tumor NOS2/COX2 proteins in DCIS shows a similar distribution of these proteins, which is consistent with earlier reports (PMID: 17096325). Importantly, unlike our spatial analyses, the TCGA database does not reflect expression at the single cell level, which thus excludes key predictive information.   The other important factors to consider are antibodies and staining conditions as well as analysis of fluorescence intensity thresholds that more accurately reflect localized protein expression in single cells as well as cell clusters that appear to be important during invasion.
